# You Can Stand Under My Umbrella: Cognitive Load in Second-Language Reading

**DOI:** 10.3390/bs15081051

**Published:** 2025-08-03

**Authors:** Francisco Rocabado, Gianna Schmitz, Jon Andoni Duñabeitia

**Affiliations:** 1Centro de Investigación Nebrija en Cognición (CINC), Department of Education, Universidad Nebrija, 28015 Madrid, Spain; jdunabeitia@nebrija.es; 2Facultat de Filología y Comunicació, Universitat de Barcelona, 08007 Barcelona, Spain; gianna.schmitz@icloud.com

**Keywords:** cognitive disfluency, second-language processing, bilingualism, virtual reality, disfluency

## Abstract

Second-language (L2) written processing has often been linked to cognitive disfluency, resembling fluency disruptions caused by perceptual challenges, such as visual degradation. This study used Virtual Reality to investigate whether cognitive disfluency in L2 mirrors perceptual disfluency by simulating adverse weather conditions (sunny vs. rainy) and applying visual masking. Spanish–English bilinguals completed a language decision task, identifying orthotactically unmarked words as either Spanish (L1) or English (L2) while experiencing these perceptual manipulations. Results showed that visual masking significantly increased reaction times, particularly for L1 words, suggesting that masking can diminish the native language advantage. Spanish words under masking elicited slower responses than unmasked ones, whereas L2 word recognition remained comparatively stable. Additionally, rainy weather conditions consistently slowed responses across both languages, indicating a general effect of environmental disfluency. A significant interaction between language and masking emerged, highlighting distinct cognitive effects for different disfluency types. These findings suggest that cognitive disfluency in L2 does not equate to perceptual disfluency; each affects processing differently. The use of Virtual Reality enabled the controlled manipulation of realistic environmental variables, offering valuable insights into how perceptual and linguistic challenges jointly influence bilingual language processing.

## 1. Introduction

Language comprehension does not occur in a vacuum. In our daily lives, we read and understand language while simultaneously navigating dynamic environments filled with sensory information that may compete for attention or interfere with processing. In bilingual contexts, these demands are especially relevant. Second-language (L2) reading is generally slower and more effortful than reading in the native language (L1), often involving additional cognitive load (e.g., greater working memory and attentional resources to support lexical access and semantic integration; ([40]; [52]; [56]). This subjective sense of effort is often referred to as cognitive disfluency, a term used to describe the feeling that a task is mentally demanding or difficult to process ([2]; [26]; [38]).

In the last 30 years, the study of multilingual language processing has continued to gain momentum, particularly as the number of multilinguals in the world’s population has rapidly grown. Research on language processing has aimed to elucidate whether there is a cognitive cost associated with processing an L2 in order to better understand the functional architecture of the bilingual mind. It is now well established that learning a new language leads to structural and functional brain adaptations as the cognitive system adjusts to managing multiple languages ([13]). This line of research has also suggested that the cognitive disfluency produced during L2 processing may, under certain conditions, mirror perceptual disfluencies caused by challenging visual input. Advances in technology, such as Virtual Reality, now enable us to examine these parallels in unprecedented ways, inviting us to reflect on past findings while relying on innovative tools to understand bilingual cognitive processing beyond previous limitations.

In the mid-twentieth century, [9] ([9]) suggested that language processing in an L2 might not be as rapid as in a multilingual’s first language. This idea is supported by more recent research, showing that L2 processing is generally less efficient than L1 processing (e.g., [55]). This difficulty becomes particularly evident in reading tasks, where increased fixation durations and a greater number of fixations have been observed during L2 reading compared to L1 reading ([21]). The authors attributed this effect to different reading strategies that imply a heavier cognitive load. This finding is in line with earlier research stating that reading in L2 takes longer than in L1, even for highly proficient readers (e.g., [11], [10]). Such increased processing demand has been linked to the simultaneous activation of both L2 and L1 representations during reading ([7]; [28]), even in contexts explicitly designed to emphasize one language (e.g., [34]; [59]; [64]).

Different explanatory models have been proposed to understand these dynamics. For instance, the Bilingual Interactive Activation plus (BIA+) model, by [14] ([14]), suggests that upon reading a word, lexical activation is automatically triggered in every language known by the multilingual. Such involuntary interference by the L1 therefore hinders the reader during the recognition of L2 words. Furthermore, the BIMOLA model proposed by [33] ([33]) pivots on the idea of two different language networks in a bilingual brain and two modes of language activation: the monolingual mode in which the base language is highly activated while the guest language network is only weakly activated, and a bilingual language mode in which both language networks are strongly activated.

This heightened mental effort observed in L2 processing aligns with the broader construct of cognitive disfluency, conceived as the subjective experience of difficulty during mental processing ([38]). According to dual-process theories of reasoning ([26]; [25]), disfluency tends to shift processing from intuitive, rapid “System 1” reasoning to slower, more effortful “System 2” reasoning. [2] ([2]) suggested that certain types of degraded perceptual input trigger such shifts. By analogy, several studies have proposed that L2 processing also incurs higher processing costs and can reduce fluency (see [12] and [62] for review), thereby qualifying as a form of cognitive disfluency. This perspective is further supported by [50]’s ([50]) model, which distinguishes between cognitive fluency (i.e., the efficiency of underlying cognitive processes) and verbal fluency (i.e., the observable output). According to this model, limitations in cognitive fluency, such as increased processing demands in L2, can underlie difficulties in verbal fluency commonly observed in non-native language use. Thus, cognitive disfluency may be a core mechanism contributing to the effortful nature of L2 processing, especially under challenging perceptual or contextual conditions.

In parallel, perceptual disfluency, understood as a subtype of cognitive disfluency induced by visually degraded stimuli, has been shown to alter judgment and processing. For instance, [39] ([39]) demonstrated that clearly visible items produce more accurate responses than perceptually degraded ones. In laboratory settings, perceptual disfluency is typically induced through manipulations such as altering fonts ([2]), blurring stimuli ([65]), or covering the items with a pattern mask ([24]; [35]; [36]).

Recently, the importance of more naturalistic methods for investigations into the processing of language has reached more prominence ([22]). Advances in technology, such as the progressive generalization of the use of Virtual Reality (VR), have provided the opportunity to merge perceptually rich experiences and close-to-real contexts with a controlled setting for experiments ([18]; [53]). VR is an ecologically valid tool for creating immersive virtual settings that can help researchers gain a deeper understanding of human cognition ([18]; [23]; [44]), facilitating exploration of the interplay between environmental context and cognition ([53]; [46]; [57]), which is otherwise difficult to achieve in a laboratory setting. In this line, recent research has turned to immersive virtual environments to simulate naturalistic disfluency. For instance, [47] ([47]) investigated the impact of visual conditions on reading performance, showing that rainy weather in VR increases fixation rates and reading times, whereas sunny weather facilitates reading. Similar claims were also raised by [45] ([45]) in a study that examined emotional valence evaluation under different weather simulations, finding that rainy conditions modestly prolonged response times but did not alter the perceived emotional valence of words. Collectively, these studies provide clear support that simulated weather can reliably serve as an ecologically valid disfluency manipulation (see [43], for recent evidence).

Importantly, from a theoretical standpoint in the field of second-language acquisition, [7] ([7]) referred to the L2 as a “degraded channel” in contrast to the L1, which is conceived as a “clear channel”, thus suggesting that L2 processing might be governed by some form of perceptual disfluency (see also [51]). Furthermore, if they are not, this discrepancy may reflect deeper differences between lab-based and ecologically embedded stimuli. Thus, our study seeks to answer whether these two disfluency types produce equivalent effects and whether they interact with the disfluency associated with L2 processing.

To address these questions, in the present study, we employed a language decision task in a VR setting to compare naturalistic and artificial disfluency inducers. This task was chosen for its sensitivity to lexical and sub-lexical processing demands, especially when using orthotactically unmarked words ([30]). Unlike lexical decision tasks, language decision tasks require not just recognizing a word but also identifying the language it belongs to. This is an ability that hinges on successful integration of phonotactic, orthotactic, and semantic cues. Previous research has shown that such tasks can detect subtle effects of cognitive load and language interference (e.g., [17]; [32]), making it a theoretically motivated tool for testing cognitive disfluency in bilinguals.

Perceptual disfluency was created through the inclusion of a rainy weather condition, following [45] ([45]) and [47] ([47]). Additionally, to test this naturalistic method in comparison with a typical laboratory setting, the current study also employed a visual mask comprising static Gaussian visual noise superimposed on the text. This method is inspired by traditional masking techniques used in perceptual research, where static or patterned visual noise is superimposed on stimuli to hinder early perceptual processing (e.g., [24] and [54] for recent examples). The rainy condition degrades the visibility of the linguistic stimuli, and it was expected to serve as a naturalistic alternative to the use of masks in laboratory settings. Participants were Spanish–English bilinguals presented with orthotactically unmarked words in both languages and asked to decide which language the word belonged to. In this context, we examined whether the naturalistic tool of virtually simulated rain would yield similar results as the laboratory tool of a visual mask. Additionally, we explored whether there is a possible similarity between the effects of perceptual disfluency and the so-called L2 processing disfluency in a way that the latter can hence be considered as real type of disfluency. Moreover, the current study investigated whether there is an interconnection between the different phenomena or if they function independently of each other. We hypothesized that if both types of perceptual disfluency—masking and rain—function similarly to L2-induced disfluency, their effects might be additive or even interactive, compounding the cognitive load. Alternatively, differences in their mechanisms might lead to dissociable patterns of interference.

## 2. Methods

### 2.1. Participants

A total of 37 students and employees from Nebrija University participated in the experiment for a monetary reward, all of them being Spanish L1 users and English L2 users. A minimum sample size of 28 participants was estimated using G*Power 3.1. ([19]) to achieve a medium effect size (α = 0.05; 1-ß = 0.95) with the current study design. Participants’ English proficiency level was assessed using LexTALE ([31]). The average level of proficiency (M = 69.72, SD = 11.44) was equivalent to a B2 level on the European Common Framework of Reference for Languages. Twenty-four of these participants self-identified as female (mean age = 21.25, SD = 3.31) and thirteen participants self-identified as male (mean age = 25.00, SD = 6.07). All had normal or corrected-to-normal visual acuity and hearing, and none reported any form of cognitive dysfunction, assessed with a computerized cognitive battery (CogniFit Inc., San Francisco, CA, USA). Participants granted written informed consent before the experiment, and the experimental procedures were approved by the Research Ethics Committee from Nebrija University (approval code UNNE-2022-0017).

### 2.2. Materials

Two hundred and forty words were used as stimuli. Half of them were Spanish words taken from the EsPal Database ([15]) and the other half were English words taken from the English Lexicon Project ([5]). To ensure task ambiguity and discourage heuristic-based decisions, only orthotactically unmarked words were included (i.e., words with valid bigram sequences in both languages). For instance, words like “shine” or “back” were excluded due to the presence of “sh” or “ck,” which are not legal bigrams in Spanish (see [8]). Additionally, words containing letters that do not exist in the other language were also excluded (e.g., niña [girl], due to “ñ”). In addition, none of the words was a perfect cognate, given that this would obviously interfere with the language decision task.

The word stimuli were divided into two lists of 120 words each (60 Spanish, 60 English), one for each weather condition (sunny and rainy). The lists were matched across key lexical variables: word frequency (Zipf scale), word length (letters), orthographic neighborhood (OLD20), within-language bigram frequency, and between-language bigram frequency (see Table 1).

### 2.3. Virtual Reality Setting and Apparatus

The setting of the experiment was created using VR via a head-mounted display (HMD). The items were presented in a 3D open street residential neighborhood that served as the main scenario due to its high quality of realism and the familiarity and openness of the simulated space. In this environment, the participants could experience simulated weather in a more realistic context (see Figure 1 and Appendix A). Model editions and 3D model implementation to the main environment were made with the Vizard inspector ([63]). It was used to remove redundant 3D objects and to integrate a white canvas for the presentation of the experimental materials. Lastly, ambient sounds were added to the VR setting to improve the immersiveness in both weather conditions. This entailed rain sounds for the rainy condition and sounds of a fountain and pigeons for the sunny condition. Additionally, the background sky was animated to comply with both weather conditions (see Appendix A for a video demonstration). Finally, to induce visual masking, a static Gaussian visual noise mask was superimposed on the white canvas in the appropriate trials. The use of visual noise followed standard methods for perceptual degradation and cognitive disfluency manipulations (e.g., [54]).

Python 2.7 ([61]) and Vizard 6 ([63]) were used to program and design the VR task. The 3D environments, including all experiment-related content, were displayed through an HTC VIVE Pro HMD at a rendering resolution of 2880 × 1600 pixels (1440 × 1600 pixels per eye). The built-in display of the headset provides a 90 Hz refresh rate as well as a 110° field of view. Participants’ viewpoint was continuously anchored throughout the experiment regardless of changes in their position in the real world.

### 2.4. Task and Procedure

Participants were provided with the HMD while seated on a rotating chair to immerse them in the abovementioned 3D virtual setting, enabling a full 360° view from a stationary perspective. After the placement and calibration of the headset, participants were equipped with two controllers, simulating two hands in the virtual setting. They were then presented with the instructions for the language decision task on a floating canvas. The instructions informed the participants of the two stages of the experiment: a practice phase followed by an experimental phase. Participants were asked to assign the linguistic stimuli to either Spanish, by pressing the button on the left controller, or English, by pressing that on the right controller. They were advised to react to the items as fast as they could and, hence, to rely on initial impressions. The target items were displayed centrally in black Courier New monospaced font on a simulated white canvas, ensuring readability. Each trial started with a central fixation cross presented for 500 ms, immediately followed by the target string, which remained visible for a maximum of 3000 ms or until a response was registered (see Figure 2). Subsequently, an inter-stimulus interval blank space was visible for 500 ms before the next trial began.

Participants started in one of the weather conditions (rainy or sunny), which was randomly assigned, and completed all 120 trials from the first list of words, comprising both Spanish and English words, with or without a visual mask superimposed (i.e., 30 items per condition in each weather context). After completion of the first block corresponding to one weather context, they were granted a 5 min break, and they were subsequently presented with the other weather conditions with a new set of 120 words.

## 3. Results

Data were preprocessed and cleaned using R 4.3 ([42]) within the RStudio environment ([49]). Reaction times (RTs) that fell outside ±2.5 standard deviations from the mean RT per participant and condition were excluded, following standard practice in psycholinguistics ([4]; [41]). This method aims to remove extreme responses, whether too fast (e.g., anticipations) or too slow (e.g., lapses in attention), without biasing results toward fixed thresholds. As a result, 3.21% (n = 136) of trials were removed in the rainy condition and 3.12% (n = 133) in the sunny condition. All excluded outliers were from the upper tail of the RT distribution. Additionally, trials with incorrect or missing responses were excluded from the RT analyses, yielding 4098 and 4127 valid observations for rainy and sunny weather conditions, respectively. The percentage (%) of correct responses per participant during the language decision task was determined as accuracy. Exploratory analyses revealed that the estimated average likelihood of accuracy rates for words was nearly at ceiling and very similar across conditions (see Table 2). Due to this, only reaction time data were analyzed.

Linear mixed-effects models were used to analyze RT data in Jamovi 2.3 ([58]), using the GAMLj module ([20]). The model included language (Spanish, English), mask (masked, unmasked), and weather (rainy, sunny) as fixed effects, along with all two- and three-way interactions. Random intercepts were included for both participants and items. The model formula (R notation) was as follows: RT ~ Language × Mask × Weather + (1|Subject) + (1|Item). Initial models tested all possible two- and three-way interactions and more complex random structures; however, their inclusion did not substantially improve model fit, as assessed by Akaike Information Criterion (AIC) comparisons. Nevertheless, their inclusion did not result in a significant detriment to model performance. For this reason, the full factorial model was retained and reported to ensure transparency and theoretical completeness. Importantly, more parsimonious models yielded qualitatively equivalent results, reinforcing the robustness of the findings.

A significant main effect of mask was observed, *F*(1, 224) = 32.37, *p* < 0.001, indicating that masked stimuli elicited slower responses than unmasked ones (21 ms difference). The weather effect was also significant, *F*(1, 8030) = 18.66, *p* < 0.001, with slower responses under rainy conditions (14 ms difference). No significant main effect of language was found, *F*(1, 224) = 0.55, *p* = 0.459.

The interaction between language and mask was significant, *F*(1, 224) = 10.01, *p* = 0.002 (see Figure 3). Post hoc comparisons (Bonferroni corrected) revealed that Spanish masked words elicited significantly slower responses than Spanish unmasked words (33 ms difference), *t*(226) = 6.25, *p* < 0.001. In contrast, no significant difference was found between English masked and unmasked words (*t*(223) = 1.79, *p* = 0.449). While Spanish and English masked words did not differ (*t*(225) = 1.71, *p* = 0.529), Spanish unmasked words were processed faster than English unmasked ones (15 ms difference), *t*(224) = −2.76, *p* = 0.037.

The interactions between language and weather, *F*(1, 8027) = 0.002, *p* = 0.963, mask and weather, *F*(1, 8026) = 0.029, *p* = 0.864, and the three-way interaction language and mask and weather, *F*(1, 8027) = 0.003, *p* = 0.958, were not significant.

## 4. Discussion

The current study aimed to explore the differences in cognitive processing elicited by the almost simultaneous retrieval of information from both the L1 and L2, as well as by simulated real-world perceptual disfluency conditions—operationalized through VR-induced meteorological scenarios and visual stimuli degradation via masking. Specifically, this study sought to determine the extent to which L2 processing might be conceptualized as a form of cognitive disfluency, comparable to that elicited by physical distortions typically used to manipulate processing fluency. To this end, we examined how visual challenges, whether derived from naturalistic weather simulations or artificial laboratory masks, affect cognitive load during a bilingual language decision task. By leveraging the immersive capabilities of VR, the present research aimed to provide a more ecologically valid framework for investigating the interplay between environmental context and bilingual language processing. In doing so, it revisited foundational questions in bilingual cognition through the lens of a naturalistic and controlled methodology that more closely reflects real-world language use.

Contrary to the initial hypothesis that L2 reading would inherently require more cognitive effort, similar to a form of perceptual disfluency, no significant main effect was found for language on response times in the language decision task. The additional cost associated with L2 reading was exclusively observed under conditions in which the target items were not visually distorted by masking. Put differently, the language effect exclusively emerged in conditions of intact presentation of the stimuli, in line with preceding reports of such an effect. In other words, L2 processing only required more cognitive effort than L1 processing when items were presented without a pattern mask. These findings nuance prior evidence linking L2 reading with higher processing demands ([10]; [28]) by showing that such effects may be contingent on stimulus clarity. When fluent processing is disrupted by perceptual manipulation, such as visual masking, the cognitive load typically associated with L2 processing appears less distinct.

The significant effects of visual masking, which led to longer response times compared to unmasked conditions, are consistent with research on perceptual disfluency ([2]) and reinforce the idea that visually degraded stimuli require greater cognitive effort. Similarly, the simulated rainy weather condition resulted in longer response times than the sunny condition (see [47]; [45]), supporting the notion that environmental manipulations in VR can effectively simulate real-world perceptual disfluency. Importantly, while weather conditions had a general effect on performance regardless of language, the masking condition revealed an interaction with language, indicating differential effects. These results suggest that perceptual disfluency, whether naturally or artificially induced, increases cognitive load ([39]) but not uniformly across contexts or linguistic conditions. A complementary, though tentative, explanation is that individuals may be more perceptually adapted to the visual irregularities introduced by natural phenomena like rainfall, which could explain why its effects appear to be more uniform and less tied to language-specific processing.

Critically, the significant interaction between language and mask indicates that different forms of disfluency operate at distinct levels, with unique impacts on language processing. Specifically, masked Spanish words led to longer response times than masked English words. The effect of masking increased response times by approximately 33 ms in Spanish and by a negligible time of 9 ms in English, suggesting that the perceptual difficulty introduced by masking overrides linguistic advantages, effectively equalizing the processing demands of L1 and L2 and mainly affecting the native language. This increased perceptual difficulty may necessitate the engagement of System 2 processing—a more deliberate and effortful cognitive system ([26])—for both languages, thereby diminishing the typical advantages of L1 processing under masked conditions. Note that System 2 is expected to be the system operating in L2 processing as a default, and thus it comes as no surprise that the only language sensitive to a system change is L1. Moreover, the presence of a language and mask interaction suggests that participants were not merely adopting an “L1/not-L1” decisional heuristic. If decisions had relied on a strategy of detecting L1 membership and rejecting everything else as “not L1”, the language-specific modulation of the visual masking would not have been expected. Supporting this, previous findings by Rocabado et al. ([47]) using a monolingual lexical decision task showed that weather-induced disfluency had no significant effects on pseudowords, which would represent the stimuli most compatible with a “not-L1” strategy. In sharp contrast, the effects on word processing were robust. Thus, a “not-L1” decisional criterion would have predicted a reduced and possibly negligible weather effect with L2 words in the current study, and the results do not support such a view.

In contrast, the effect of rain was additive and did not interact with language. Rainy conditions increased response times by approximately 12 ms in Spanish and 15 ms in English. These results replicate our previous findings and confirm that naturalistic visual disfluency affects L1 and L2 similarly. The lack of interaction indicates that environmental disfluency of this type imposes a general increase in cognitive load rather than modulating language-specific processing (see [43] for recent evidence of similar results). This distinction supports the broader theoretical implication that not all disfluencies operate equally: while some, like masking, selectively disrupt L1 processing, others, such as weather-related degradation, impact all linguistic input equally. Such dissociations reveal important boundaries in how perceptual and linguistic systems interact.

These results align with broader literature describing how increased perceptual difficulty and linguistic interference can differentially impact language processing under challenging conditions. For instance, [60] ([60]) showed that speech recognition is more adversely affected by background noise in the same language as the target compared to a different language, suggesting that linguistic interference from a familiar language can amplify perceptual challenges. Similarly, [37] ([37]) found that native language background noise had a stronger masking effect on memory for speech than non-native background noise, indicating that semantic interference from the native language intensifies processing demands. However, [3] ([3]) found that under conditions of severe perceptual degradation—as with cochlear implant users attempting to distinguish high-pass-filtered musical sounds—linguistic distinctions become less relevant, as extreme perceptual limitations reduced participants’ ability to discern between stimuli, regardless of language. These findings collectively suggest that while linguistic factors can indeed heighten word language recognition, conditions where the quality of the linguistic material is compromised, such as classical laboratory masking conditions where the quality of the stimuli is markedly degraded, can override these language-based effects at higher rates compared to recreating natural environmental conditions. In our study, this was evident as masking-induced perceptual disfluency reduced the typical processing advantage of the L1, effectively leveling the cognitive load across languages. This supports the idea that under highly challenging perceptual conditions, native language processing demands can approach those of the L2, illustrating the complex interplay between perceptual and linguistic factors. This reinforces the hypothesis that perceptual challenges can neutralize the usual cognitive benefits associated with L1 processing, particularly when perceptual clarity is severely compromised. In such cases, the typical cognitive cost associated with L2 processing—often attributed to disfluency in lexical access or decision-making ([50]; [12])—becomes less distinguishable, as both languages are processed under conditions that force a shift toward more effortful, System-2-like mechanism reasoning ([26]).

Beyond these empirical findings, the use of VR in this study demonstrates its potential for cognitive research. VR allows researchers to simulate ecologically valid environments while maintaining experimental control, see [48] ([48]) for a recent demonstration. Our study exemplifies how VR can be leveraged to investigate language processing in contexts that closely approximate everyday challenges. This approach aligns with growing trends in cognitive science that emphasize naturalistic paradigms and complements prior work showing how embodied, situated cognition can inform theoretical models ([47]).

The present study has several limitations that warrant consideration. First, although our design aimed to simulate realistic disfluency conditions, it focused on a relatively constrained task: language decision using unmarked words. Future research should explore whether these effects generalize to more complex linguistic materials, such as sentences, narratives, or interactional speech. Second, while the use of unmarked words reduced orthographic bias, it may also have elevated cognitive demands by requiring decisions based on higher-level lexical–semantic processing rather than perceptual cues ([30]; [16]). This may partly explain the absence of a main effect of language. Future studies could directly manipulate orthotactic markedness to examine how surface features facilitate or hinder bilingual language processing under disfluent conditions. Third, although word familiarity was not directly assessed, this variable was not deemed critical for the aims of the current study. The stimuli were deliberately selected to be lexically simple, unmarked, and visually balanced across conditions, minimizing variability related to prior exposure. Nonetheless, future studies could explicitly explore how familiarity may influence processing ease by manipulating it. Fourth, although the weather manipulation was intended to induce perceptual disfluency, previous research has shown that mood can modulate affective word processing ([1]; [6]) and that weather conditions are linked to fluctuations in emotional state ([27]; [29]). However, in a recent study using the same simulated environments, weather conditions did not significantly alter the emotional evaluation of valenced words ([45]), suggesting that emotional state is unlikely to have been a major driver in the present findings. Moreover, the randomized block design likely minimized any systematic mood differences across conditions. Even so, future studies would benefit from incorporating manipulation checks of mood or arousal to empirically disentangle emotional influences from perceptual ones. Finally, our sample size may have shown limited power to detect subtle effects. Larger samples and complementary paradigms (e.g., EEG, pupillometry) could offer a more granular understanding of how disfluency and language interact.

In conclusion, the present findings provide evidence that L2 processing, though more effortful under ideal visual conditions, becomes indistinguishable from L1 processing when perceptual disfluency is introduced. The differential effects of masking and weather highlight the importance of distinguishing between artificial and natural sources of disfluency in bilingual language processing. The use of VR enhances ecological validity and offers new avenues for studying how everyday environments modulate cognitive load. These findings contribute to both theoretical debates and methodological innovation in bilingualism research, paving the way for future work examining how perceptual, linguistic, and contextual factors jointly shape language comprehension.

## Figures and Tables

**Figure 1 behavsci-15-01051-f001:**
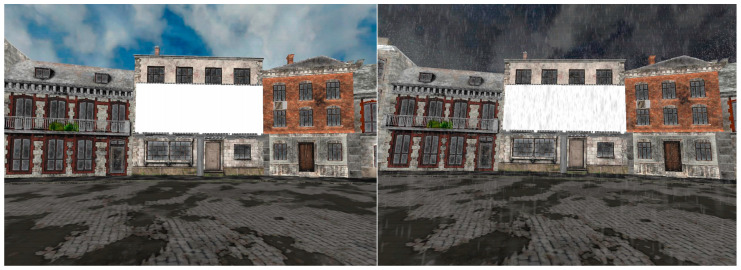
Illustration of the main scenario from the participant’s viewpoint under sunny (**left** image) and rainy (**right** image) weather conditions.

**Figure 2 behavsci-15-01051-f002:**
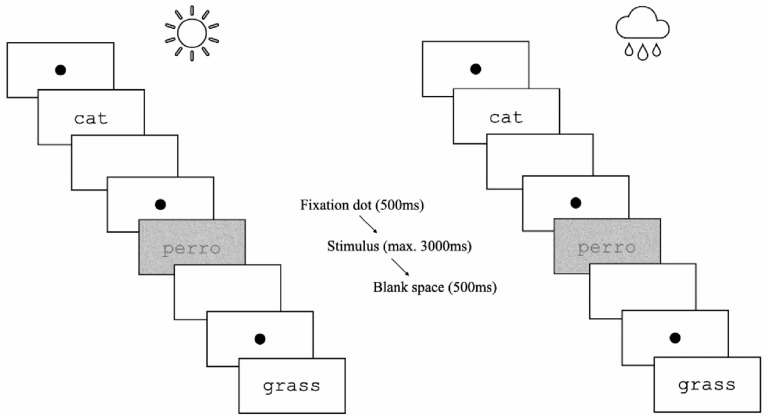
Representation of the structure of three sequential trials in the sunny and rainy blocks. Unmasked and masked conditions and English and Spanish stimuli are shown for illustration purposes.

**Figure 3 behavsci-15-01051-f003:**
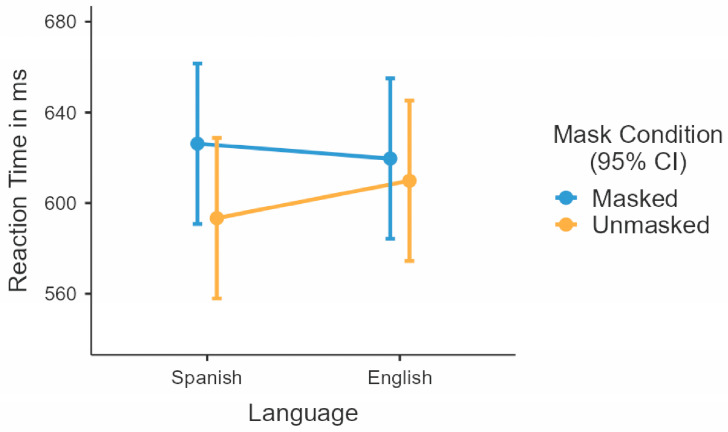
Interaction effect between language and mask in the reaction time analysis. Bars represent 95% confidence intervals.

**Table 1 behavsci-15-01051-t001:** Descriptive statistics of characteristics of the materials.

Word Properties	English		Spanish	
	List 1	List 2	List 1	List 2
Word frequency	2.56 (0.39)	2.52 (0.44)	2.55 (0.33)	2.54 (0.32)
Word length	5.62 (0.49)	5.48 (0.50)	5.48 (0.50)	5.45 (0.50)
Within-language bigram frequency	0.81 (0.33)	0.79 (0.31)	0.87 (0.28)	0.82 (0.34)
Between-language bigram frequency	0.67 (0.23)	0.72 (0.21)	0.67 (0.28)	0.63 (0.27)
Orthographic neighborhood	1.85 (0.28)	1.82 (0.25)	1.55 (0.24)	1.48 (0.24)

Values reported are means with standard deviation in parentheses for word frequency (Zipf scale), word length (number of letters), within-language bigram frequency (percentage per million), between-language bigram frequency (percentage per million), and orthographic neighborhood, measured through the average orthographic distance to the 20 nearest neighbors to indicate neighborhood density (OLD20).

**Table 2 behavsci-15-01051-t002:** Descriptive analysis of accuracy proportions and reaction times (in milliseconds) across language, mask, and weather conditions. Means are reported with standard deviations in parentheses.

Language	Mask Condition	Weather Condition	Accuracy M (SD)	Reaction Time M (SD)
**Spanish**	Masked	Rainy	0.94 (0.23)	636 (192)
		Sunny	0.95 (0.21)	623 (183)
	Unmasked	Rainy	0.96 (0.20)	599 (194)
		Sunny	0.97 (0.17)	588 (161)
**English**	Masked	Rainy	0.96 (0.19)	625 (174)
		Sunny	0.97 (0.18)	613 (151)
	Unmasked	Rainy	0.96 (0.20)	618 (171)
		Sunny	0.96 (0.20)	600 (166)

## Data Availability

Data supporting this study can be found at the following link: https://osf.io/5rgwj (accessed on 4 November 2024).

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
