# Peer review of "You Can Stand Under My Umbrella: Cognitive Load in Second-Language Reading"

_behavsci, 2025, doi:10.3390/bs15081051_

Round 1

Reviewer 1 Report

Comments and Suggestions for Authors

In appendix, thanks.

Author Response

We sincerely thank the reviewer for the time and effort they dedicated to evaluating our manuscript. Their thoughtful and constructive comments have significantly improved the clarity, rigor, and overall quality of the work.

Comments 1:    The statistical analysis is a little bit simplified. It only reports the Language and Mask interaction and doesn't explore the three-factor interaction (Mask and Weather and Language) in more detail. I suggest adding a three-factor interaction analysis to see if the effect of visual masking is different in rainy conditions. Even if there's no significant interaction, you should still provide the analysis results to make the results more transparent and discuss them.

Response 1:    We thank the reviewer for this insightful suggestion. While we initially reported the most parsimonious model based on AIC comparisons, we indeed tested a more complex model including the full three-way interaction between Mask, Weather, and Language. Although this interaction was not significant, we agree that its inclusion adds transparency and completeness to the results. We have now included the results of this full model in the manuscript, results section.

Comments 2:    The rationale for using the language decision task (LDT) requires further clarification. The article employs a language decision task (LDT) to assess additional cognitive load during second language processing; however, the results of this task did not reveal a significant main effect of language. Have other researchers in the literature also used this task to examine cognitive load? If so, please include this information in the introduction; if not, please provide a detailed analysis of why this task can be used to assess additional cognitive load or add relevant literature to explain which tasks previous researchers have used to study language cognitive load.

Response 2:    We thank the reviewer for pointing out the need to better justify our choice of the language decision task. This task is classically used to investigate language-specific activation and lexical access (e.g., Casaponsa & Duñabeitia, 2016), and in our design it also served to explore how cognitive load is modulated under ambiguity. Specifically, we selected orthotactically unmarked words—i.e., words with bigrams that are plausible in both Spanish and English—to remove early sub-lexical cues that would normally facilitate rapid language classification. As previous studies have shown (Lecerf et al., 2024), this increases the need for lexical-semantic processing, as participants cannot rely on surface features to decide the language. Consequently, reaction times become slower and differences between L1 and L2 are reduced, especially young adults (Duñabeitia et al., 2020). This design allowed us to create a demanding cognitive environment and observe how additional perceptual disfluency (e.g., visual masking or simulated rain) interacts with already effortful language processing.
We have clarified this rationale in the introduction and methods section, and added references to related studies that support the idea that when orthographic markedness is minimized, language decisions become slower and cognitively more demanding, especially in the L1 where rapid access is otherwise more likely.

Duñabeitia, J. A., Borragán, M., de Bruin, A., & Casaponsa, A. (2020). Changes in the Sensitivity to Language-Specific Orthographic Patterns With Age. Frontiers in Psychology, 11. https://doi.org/10.3389/fpsyg.2020.01691
Lecerf, M.-A., Casalis, S., & Commissaire, E. (2024). New insights into bilingual visual word recognition: State of the art on the role of orthographic markedness, its theoretical implications, and future research directions. Psychonomic Bulletin & Review, 31(3), 1032-1056. https://doi.org/10.3758/s13423-023-02347-6

Comments 3:    Although word frequency and length were controlled, word familiarity was not controlled, and word familiarity has a significant impact on L2 processing time. It is recommended that the authors clarify whether word familiarity was controlled or whether this factor was considered in the selection of materials. If it was not controlled, this should be noted in the "Limitations" section, and it is recommended that future studies include this factor.

Response 3:    Thank you for pointing this out. Although we did not directly assess subjective word familiarity, our stimuli were selected based on objective lexical properties commonly associated with familiarity, such as word frequency and orthographic simplicity. Specifically, we ensured that all items were frequent, short, and orthographically regular in both languages, minimizing potential confounds related to perceptual or decoding difficulty. We have now pointed out this in the Discussion section. Future studies could explicitly manipulate or measure subjective familiarity to further disentangle its role in bilingual word recognition.

Comments 4:    The discussion part is lacking in depth and does not provide a thorough analysis of the results. The. discussion. mentions, ""All in all, these results reflect that perceptual disfluency, whether it. comes from an artificial source or within the real- world context, increases cognitive load." The authors are requested to clarify what "artificial source" and "within the real-world context" specifically refer to (since the authors' findings indicate that weather conditions produced generalized effects that equally impacted L1 and L2 processing, while the masking conditions had a differential effect depending on the language of the words), the narrative is somewhat vague and contains contradictions. It is recommended that the authors revise the wording of the discussion part.

Response 4:    We thank the reviewer for this thoughtful observation. In the revised Discussion section, we have clarified the distinction between the two types of perceptual disfluency investigated: “artificial sources” refer to the use of visual masking—a classic laboratory manipulation that selectively degrades the visibility of stimuli—while “real-world context” refers to ecologically valid environmental manipulations, such as the simulated rain condition presented through VR.
We now explicitly state that while both manipulations increased response times, only the artificial disfluency (masking) interacted with Language, suggesting that different types of disfluency affect cognitive processing in qualitatively distinct ways. This updated narrative emphasizes that masking disproportionately impacted L1 processing, whereas rain exerted a uniform effect across languages. These clarifications resolve the ambiguity noted by the reviewer and strengthen the theoretical interpretation of the results.

Comments 5:    The discussion part suggests that the authors discuss each experimental finding separately in separate modules rather than discussing all findings together. In addition, the article lacks a Limitations and Future directions part. For example, the experiments in the article only used reaction time (RT) as an indicator to measure cognitive processing speed and load, which is relatively subjective. In the future, neurophysiological methods could be used for more in-depth investigation. The authors are advised to supplement the article with such information.

Response 5:    We agree that a clearer structure benefits the discussion. Accordingly, we have now reorganized the Discussion to better reflect our findings. Additionally, we added a new section on Limitations and Future Directions. In this section, we note that RT is a limited proxy for cognitive load and suggest using eye-tracking or pupillometry in VR settings in future research. Although VR-EEG setups have been explored, we caution against their use due to signal quality issues. These ideas are now incorporated in the final part of the Discussion.

Comments 6:    The introduction part needs improvement. The manuscript discusses disfluency in visual perception and language cognition, but the introduction does not distinguish between the processing stages or mechanisms of the two disfluency. Based on the literature analysis, it is possible to determine at which stage L2 processing may experience disfluency and whether masking may have an impact on it. It is recommended that the authors add more literature content and provide additional information.

Response 6:    We thank the reviewer for this insightful suggestion. In response, we have revised the Introduction to clearly differentiate between perceptual and cognitive disfluency, specifying their mechanisms and stages of processing. We now include a discussion based on Segalowitz’s (2010) triadic framework of fluency—comprising cognitive fluency, utterance fluency, and perceived fluency. This model helps to situate disfluency effects within specific stages of processing: perceptual disfluency, such as visual masking, primarily affects early stages of visual encoding and stimulus identification, while L2-related cognitive disfluency arises later, during lexical access, decision-making, and response selection.
By clarifying these distinctions, we aim to show how the different sources of disfluency—one rooted in perceptual input degradation and the other in higher-level linguistic demands—may converge or diverge in their impact on task performance. These revisions provide the theoretical structure necessary to interpret our findings within a multistage processing framework.

Comments 7:    The highlight of this paper is the use of VR to simulate "naturalized sensory interference," but this point is not sufficiently emphasized in the introduction and conclusion. It is suggested that the authors emphasize the core.highlights of the paper.

Response 7:    We appreciate this suggestion. The use of VR to simulate naturalistic perceptual challenges is indeed a central innovation of this study. We have now made this point more explicit both in the Introduction and Conclusion. We highlight how VR allows researchers to test perceptual-cognitive interactions in ecologically valid contexts that would otherwise be difficult to reproduce in traditional lab settings.

Reviewer 2 Report

Comments and Suggestions for Authors

Thank you for the opportunity to review your manuscript. The manuscript is well written and convincingly argued. I also appreciated the online supplement and the videos to understand your study. I mostly have minor remarks, often pertaining to consistent APA style in citations and statistical reports. However, one major point of revision is the disjunction of the results and the discussion section with regard to the reporting of the language x weather interaction that needs to be remedied before the manuscript can be approved for publication.

l40: You need to define cognitive disfluency and cognitive load a bit earlier. E.g. l40, what do you mean with that, and do you have references for that construct?

l49: double brackets (())

l51: Check APA Style for in-text citations

l66: Citation? Is Lewy part of the paper or not?

l147: It is for me somewhat difficult to understand how the mask relates to laboratory settings of previous studies. I suggest you expand on this topic briefly. 

l159: check APA style for in-text reporting of statistics (italics).

l173: "shine" or "back" for readability

l240: Given that the RT is limited to 0 and 3 seconds, I am not sure if excluding reaction times that are longer than ~1.2 seconds is an appropriate. I understand that you want to avoid rapid guessing behaviour at the lower end, but what would be a reason to exclude data where respondents answered but took longer than 1.5 seconds? I would try to justify that decision and report the percentage of exclusion on each end for the conditions. Alternatively use a method like Wise & DeMars (2006) to account for rapid guessing behaviour. 

l260: Would you have had reasons to suspect these interactions, or were the analyses exploratory in nature? From what I understand, for your research questions, both masking and rain work as stimuli that elicit extraneous cognitive load. As such, even if the models do not improve, it would still make sense to report a language x rain condition parallel to the mask x language condition to see if they are interchangeable. From your theory, I would also expect that both conditions compound on each other, so a three-way interaction would also be interesting. Currently, your results do not fully answer what you describe in ll139-141. 

l306: This is kind of a problem with the LDT, since participants do not necessarily need to decide if the presented word is L1 or L2, but they can also choose to decide if the word is L1 or not L1. However, you have some evidence that they did not do that, as then there probably would be no need to use a system 2 response on L2 words (since the process is in L1 only) and then there probably would not be an interaction effect... 

l317: Again, without the language x weather interaction in the results to compare to the language x mask condition, I do not think you can fully make that statement. Assuming they work the same, they should have the same interaction effects. If they do not, there might be a difference between the ecologically valid and the laboratory setting in some way or another. 

l338: Here you are discussing the interaction effect and thus analyses that you did not report (anymore?). I suspect I am disagreeing with a previous reviewer on the topic of reporting the interaction effect here? 

Author Response

We are very grateful to the reviewer for their insightful feedback and careful reading of our manuscript. The comments provided were instrumental in refining key aspects of the study and strengthening the final version.

Comments 1: l40: You need to define cognitive disfluency and cognitive load a bit earlier. E.g. l40, what do you mean with that, and do you have references for that construct?

Response 1: We thank the reviewer for this helpful suggestion. We have now introduced definitions of both cognitive disfluency and cognitive load earlier in the Introduction, this provides a clearer conceptual grounding for the distinctions we make later in the paper.

Comments 2 and 3:    l49: double brackets (()) ///  l51: Check APA Style for in-text citations

Response 2 and 3:    We sincerely apologize for the formatting issues. A previous version of the manuscript contained placeholder-style references that were not yet conformed to the journal’s formatting guidelines was submitted journal’s platform. In the current version, all in-text citations have been revised to adhere to the journal referencing stile.

Comments 4:    l66: Citation? Is Lewy part of the paper or not?

Response 4:    Thank you for pointing this out. In the original version of the manuscript, we cited Grosjean (2008), who provides a comprehensive overview of the BIMOLA model in a book chapter authored solely by him. However, we acknowledge that this model was originally developed in collaboration with Léwy. To properly credit both contributors, we have now added the original reference to the BIMOLA model by Léwy and Grosjean. The revised citation in the manuscript reflects this clarification.

Comments 5:    l147: It is for me somewhat difficult to understand how the mask relates to laboratory settings of previous studies. I suggest you expand on this topic briefly.

Response 5:    We appreciate this suggestion. We have now expanded the description of the masking procedure and added explicit references to earlier work on visual masking (e.g., Hirshman & Mulligan, 1991) to clearly relate our method to classic laboratory paradigms. This should help clarify the conceptual and methodological continuity between prior research and the present study.

Comments 6:    l159: check APA style for in-text reporting of statistics (italics).

Response 6:    We have reviewed and updated all statistical reports to follow the journal’s style guidelines (e.g., italicizing statistical symbols such as F, p, etc.) throughout the manuscript.

Comments 7:    l173: "shine" or "back" for readability

Response 7:    Thank you for the suggestion. We have now enclosed the examples “shine” and “back” in quotation marks for clarity.

Comments 8:    l240: Given that the RT is limited to 0 and 3 seconds, I am not sure if excluding reaction times that are longer than ~1.2 seconds is an appropriate. I understand that you want to avoid rapid guessing behaviour at the lower end, but what would be a reason to exclude data where respondents answered but took longer than 1.5 seconds? I would try to justify that decision and report the percentage of exclusion on each end for the conditions. Alternatively use a method like Wise & DeMars (2006) to account for rapid guessing behaviour.

Response 8:    We thank the reviewer for raising this important point. We followed a standard outlier trimming procedure widely used in psycholinguistic research (e.g., Baayen & Milin, 2010; Ratcliff, 1993), excluding trials whose reaction times fell outside ±2.5 standard deviations from the mean RT per participant and condition. This approach is designed to remove extreme values—whether too fast (e.g., premature responses) or too slow (e.g., attentional lapses)—without biasing the data toward a fixed duration.
In response to your helpful suggestion, we now report in the manuscript the percentage of trials excluded from both the lower and upper ends of the distribution, separately for each weather condition. This should provide a clearer view of how the trimming process impacted the dataset while preserving the generalizability of our findings.

Comments 9:    l260: Would you have had reasons to suspect these interactions, or were the analyses exploratory in nature? From what I understand, for your research questions, both masking and rain work as stimuli that elicit extraneous cognitive load. As such, even if the models do not improve, it would still make sense to report a language x rain condition parallel to the mask x language condition to see if they are interchangeable. From your theory, I would also expect that both conditions compound on each other, so a three-way interaction would also be interesting. Currently, your results do not fully answer what you describe in ll139-141.

Response 9:    Thank you for this insightful question. Our study was somewhat exploratory yet theoretically grounded in the assumption that both visual masking and simulated weather conditions impose extraneous cognitive load, and that such load might either affect processing additively or interactively.
Our original rationale for reporting a reduced model was based on statistical parsimony: using AIC comparisons, we selected the model that best balanced fit and complexity. The three-way interaction (Language and Mask and Weather) and the two-way interactions involving Weather (i.e., Language and Weather, Mask and Weather) were not significant and did not improve model fit. From this perspective, we initially opted not to retain or report them.
However, we fully agree with your assessment that these interactions are theoretically relevant. Indeed, our study was designed around the hypothesis that both masking and simulated weather impose perceptual disfluency and might function similarly—or perhaps compound—in their effects on cognitive processing. As such, testing whether these two forms of disfluency interact with language in comparable or distinct ways is crucial for evaluating whether they operate through shared or separate mechanisms. We also appreciate that this issue connects directly with the theoretical point we raised in lines 139–141 regarding the potential interchangeability or additivity of perceptual and linguistic disfluency.
Taking this into account, we have updated our analysis to include the full factorial model, incorporating all two- and three-way interactions. While only the Language and Mask interaction reached statistical significance, the presence of non-significant interactions is itself informative: the Language and Weather interaction was not significant, suggesting that naturalistic disfluency (rain) had a general effect on processing, affecting both L1 and L2 similarly. Similarly, the three-way interaction was also non-significant, indicating no compounding effect between both types of disfluency.
These results, now reported in the revised manuscript, support the interpretation that artificial and naturalistic disfluency both increase cognitive load, but do so in functionally distinct ways: masking appears to disrupt L1 processing disproportionately (possibly by interfering with early sublexical access), whereas weather introduces a more global perceptual challenge that affects both languages equally. We believe this more complete reporting now directly addresses your theoretical concern and clarifies the mechanisms involved.

Comments 10:    l306: This is kind of a problem with the LDT, since participants do not necessarily need to decide if the presented word is L1 or L2, but they can also choose to decide if the word is L1 or not L1. However, you have some evidence that they did not do that, as then there probably would be no need to use a system 2 response on L2 words (since the process is in L1 only) and then there probably would not be an interaction effect...

Response 10:    We appreciate this insightful observation. Indeed, in tasks involving bilingual lexical decision, participants might theoretically adopt a simplified L1/not-L1 decision strategy—particularly if sublexical or familiarity cues allow for rapid exclusion of non-native words.
However, several aspects of both our current and prior work suggest that this strategy was not operative. First, our stimulus set consisted exclusively of orthotactically unmarked words, minimizing early sublexical cues that could help participants reject stimuli as non-L1 based on orthographic patterns. This design forces participants to engage in deeper lexical-semantic processing, regardless of language membership.
Second—and most critically—we previously examined a similar scenario in Rocabado et al., 2024 (Experiment 1), where participants performed a lexical decision task (words vs. pseudowords) under simulated weather conditions. That paradigm mimics the logic of an L1/not-L1 decision, since pseudowords functionally represent “non-L1” stimuli. If participants in that study were using an L1-based shortcut, we would expect weather-induced disfluency to affect only L1 word decisions, but not pseudoword rejections. However, the results showed that weather effects were not significant for pseudowords—despite these being the best candidates for a heuristic-driven rejection. This suggests that participants did not consistently apply an L1/not-L1 simplification, but instead processed both types of items under deeper evaluative mechanisms.
Finally, the presence of a significant Language and Mask interaction in the current experiment—where L1 responses were disproportionately slowed under masking—further challenges the L1/not-L1 shortcut explanation. If participants were collapsing all unfamiliar items into a general "non-L1" category, we would expect this interaction to be absent. Instead, the observed language-specific effects imply that participants engaged in genuine language discrimination and not mere exclusion.
We have clarified this rationale in the revised Discussion, reinforcing the interpretability of the current LDT paradigm and aligning it with our prior findings.

Comments 11:    l317: Again, without the language x weather interaction in the results to compare to the language x mask condition, I do not think you can fully make that statement. Assuming they work the same, they should have the same interaction effects. If they do not, there might be a difference between the ecologically valid and the laboratory setting in some way or another.

Response 11:    Your comment rightfully points out that, for a valid comparison between the effects of masking and simulated weather, both should be tested in interaction with Language. The updated results confirm that the interaction between Language and Mask is significant, whereas the interaction between Language and Weather is not. This suggests that masking and simulated weather do not operate in the same way: while masking introduces differential costs that affect L1 more strongly, simulated weather appears to exert a more uniform perceptual load on both languages.
We have revised the Discussion accordingly to reflect this dissociation, and to avoid implying equivalence between the two manipulations in the absence of parallel interaction effects.

Comments 12:    l338: Here you are discussing the interaction effect and thus analyses that you did not report (anymore?). I suspect I am disagreeing with a previous reviewer on the topic of reporting the interaction effect here?. 

Response 12:    Thank you for pointing this out. To clarify, the sentence you are referring to was originally based on patterns observed in the descriptive statistics (i.e., mean differences between rainy and sunny conditions within each language) rather than on inferential tests of the Language and Weather interaction. The numerical differences (12 ms for Spanish, 15 ms for English) reflected average RT increases across weather conditions and were used to illustrate the apparent similarity in how rain affected both languages. However, at that stage, no formal statistical interaction test had been conducted for this comparison.
There was no previous round of review in which the interaction was removed—this manuscript is being reviewed for the first time. Nevertheless, we acknowledge that interpreting this difference without testing it statistically was potentially misleading, especially in light of the importance of this interaction for the theoretical interpretation of the results.
The manuscript has been revised to clarify that this conclusion is now based on inferential statistics, and the descriptive values are reported as illustrative only.
We have also adjusted the wording in the Discussion to eliminate any ambiguity and to ensure consistency between the reported results and their interpretation.

Round 2

Reviewer 1 Report

Comments and Suggestions for Authors

A key concern remains before I can recommend acceptance: Does rainy or sunny weather elicit different emotional states in participants? If so, it becomes critical to determine whether these induced states or cognitive disfluency is the primary factor influencing word processing. Manipulation tests may be helpful to resolve this issue. 

Author Response

Thank you once again for your valuable feedback and for your careful consideration of our manuscript during this second round of review. We truly appreciate the time and thought you have dedicated to improving our work.

We hope these changes adequately respond to your comments and contribute to the clarity and robustness of the manuscript.

Reviewer comment:
A key concern remains before I can recommend acceptance: Does rainy or sunny weather elicit different emotional states in participants? If so, it becomes critical to determine whether these induced states or cognitive disfluency is the primary factor influencing word processing. Manipulation tests may be helpful to resolve this issue.

Response:
Thank you for raising this important point. We have now addressed this issue in the revised manuscript by adding a dedicated paragraph to the Limitations section (see lines in blue). While prior studies have shown that mood can influence affective word processing and that weather conditions can modulate emotional states, a recent study using the same virtual weather environments as the current experiment (Rocabado & Duñabeitia, 2024) found no significant effects of weather on the emotional evaluation of valenced words. These findings suggest that mood is unlikely to have been a major factor in the present results. Additionally, the randomized block design used in our study mitigates systematic mood effects across conditions. Nonetheless, we acknowledge that future studies would benefit from incorporating direct measures of mood or arousal to empirically dissociate emotional from perceptual contributions to language processing.

Reviewer 2 Report

Comments and Suggestions for Authors

Thank you for the revised version of the manuscript and for addressing the criticism brought forward. 

Author Response

Dear Reviewer,

Thank you very much for your kind words and for your time reviewing the revised version of our manuscript. We truly appreciate your thoughtful feedback throughout the process. We are glad to know that the revisions addressed the concerns raised.

With best regards,
Francisco Rocabado and Jon Andoni Duñabeitia